# Factors affecting online health information-seeking behavior in young and middle-aged patients with stroke

Ge Shi[1‡], Jiajia Yu[2‡], Jiaming Zhang[1], Jun Zhao[1], Zhen Peng[1], Li Shang[2]*

**1** School of Nursing (School of Gerontology), Binzhou Medical University, Yantai, Shandong, China,
**2** Yantai Affiliated Hospital of Binzhou Medical University, Yantai, Shandong, China

‡ GS and JY also contributed equally to this work.
* shangli1977@163.com

## Abstract

This study aimed to explore the characteristics of online health information-seeking behavior and the influencing factors among young and middle-aged Chinese patients with stroke. The participants of this study were 230 young and middle-aged patients with stroke enrolled from a Class III Grade A hospital in Shandong Province, China, using convenience sampling from October 31, 2023, to May 15, 2024. Based on relevant theories and literature reviews, a self-administered questionnaire was used to analyze the influencing factors regarding six aspects: general demographic characteristics, disease factors, psychological factors, environmental factors, information factors, and information technology factors. Univariate, Correlation, and multivariate analyses were conducted to explore the factors affecting online health information-seeking behavior. The results showed that age, literacy level, stroke course, hospitalizations, treatment methods, number of combined chronic diseases, perceived usefulness, perceived ease of use, e-health literacy, self-efficacy, perceived benefit, health anxiety, quality of information, social influence, perceived risk, and privacy of information were all factors that influenced the online health information-seeking behavior in young and middle-aged patients with stroke. Age, perceived risk, and information privacy were negatively associated with online health information-seeking behavior, whereas the other variables were positively correlated. This study provides scientific insights into the intervention of online health information-seeking behavior in young and middle-aged patients with stroke and contributes to the enhancement of online health information literacy.

## Introduction

Stroke, a common neurological disorder, causes high morbidity and mortality rates worldwide. In 2022, the World Stroke Organization (WSO) reported more than 12.22 million new stroke cases worldwide each year, and the number of people who die

**Data availability statement:** All relevant data are within the manuscript and its Supporting Information files.

**Funding:** This study was supported by the fund for teaching reform and research of Binzhou Medical University (Grant No. 2019YD060).The project is named "The Effect of Community-Family Precision Care with the Involvement of Primary Caregivers on Stroke Hemiplegia Patients after Discharge" and the fund was awarded to the co-author Jiajia Yu of this article.

**Competing interests:** The authors have declared that no competing interests exist.

from stroke is approximately 6.55 million [1]. Stroke has become the main cause of death and disability worldwide, posing a major threat to human health [2]. Studies have shown that from 1990 to 2021, the stroke incidence rate among people under 70 years of age increased by 14.8% [3]. Globally, approximately 10–20% of stroke events occur in young people [4]. The age of stroke onset has gradually decreased. Young and middle-aged individuals are in ascending stages of career development and have significant family responsibilities. When young and middle-aged people have stroke, it not only seriously affects their daily activities and self-care ability, but also poses burdens on their families [5]. Thus, stroke has a significant negative impact on young and middle-aged people, who are the main labor force.

Good health information-seeking behavior is beneficial for patients to understand diseases and enhance disease management abilities, thereby enhancing rehabilitation effects and reducing disease incidence rates [6]. As an important source of health-related information, the Internet has demonstrated a significant increase in the number of users [7]. In the United States, an increasing number of Americans obtain health and medical information through the Internet. Approximately three-quarters of all individuals search for health information online [8]. A study by the European Union found that research on online health-related information behavior has received attention in multiple countries and regions. Taking Finland as an example, in 2017, among Finnish people aged 16–89 years, 64% searched for health-related information online [9]. People regard the Internet as a way of obtaining health information, and the process of obtaining health information on the Internet is known as online health information-seeking behavior (OHISB) [10].

However, some problems are present in the OHISB of young and middle-aged patients with stroke, such as low information quality, excessive information volume [11], and low e-health literacy [12]. These problems can lead to misleading adverse effects in young and middle-aged patients with stroke and can even cause harm to their physical health. Therefore, studying the factors influencing OHISB in young and middle-aged patients with stroke is of great significance for improving OHISB and enhancing self-management abilities and rehabilitation effects.

This study aimed to understand the current status of OHISB in young and middle-aged patients with stroke. Additionally, based on relevant theories and a literature review, we aimed to comprehensively and systematically explore the influencing factors. We analyzed the factors influencing OHISB in young and middle-aged patients with stroke regarding six aspects: general demographic characteristics, disease factors, psychological factors, environmental factors, information factors, and information technology factors to provide a reference for promoting OHISB in young and middle-aged patients with stroke.

## Theoretical development and hypotheses

### The second Wilson model

Wilson proposed the first information behavior model in 1981 [13]. This model is user-centered and considers complete information-seeking behavior as a unit but ignores the effects of situational factors on information-seeking behavior. In 1996,

Wilson improved this model [14]. The new model considers that the sought information by users in different situations will be different, and the information behavior will be generated through the influence of incentive mechanisms and intervention variables. In addition, information behavior will give information feedback to users, again affecting subsequent information needs and behaviors, forming a circular mechanism. First, the theoretical model is comprehensive and includes information-seeking behavior, intervention variables (such as source characteristics), and activation mechanisms (such as self-efficacy). Second, it is in line with the purpose of this study, which is to investigate the influencing factors of OHISB in young and middle-aged patients with stroke who have a demand for health information. Therefore, Wilson's general information behavior model was used as the theoretical framework to provide a theoretical reference for this study. Because the model contains many variables, this study did not aim to test the second Wilson model but to examine the relationship between the intervention variables (demographic factors, situational factors, information source characteristics, social roles, and psychological factors), activation mechanisms (self-efficacy), and OHISB.

### Research hypotheses

According to the theoretical model, the influencing factors were divided into general demographic characteristics as well as disease, psychological, environmental, information, and information technology factors.

**General demographic characteristics.** Sex is a prominent personal characteristic. Young women and men pay different amounts of attention to health information. This is related to differences in their physiological characteristics, circles of friends, family conditions, and social environment [15]. A study on patients with thyroid cancer found that age and education level were also significant predictors of digital divide [16], In general, younger and more educated individuals receive more mobile intelligence training and have higher Internet self-efficacy, which may affect OHISB. A study on Lithuanian patients found that differences in monthly household incomes and occupations were also significant predictors of digital divide [17]. Accordingly, the following hypotheses were proposed in this study:

H1: Sex is an influencing factor for OHISB in young and middle-aged patients with stroke.

H2: Age is an influencing factor for OHISB in young and middle-aged patients with stroke.

H3: Education level is an influencing factor for OHISB in young and middle-aged patients with stroke.

H4: Monthly income is an influential factor for OHISB in young and middle-aged patients with stroke.

H5: Occupational difference is an influential factor for OHISB in middle-aged and young patients with stroke.

**Disease factors.** Based on the actual clinical situation combined with the literature review, indicators that could represent the disease information in young and middle-aged patients with stroke were selected, including stroke course, hospitalizations, treatment methods, and number of combined chronic diseases. Therefore, we proposed the following hypotheses:

H6: Stroke course has a significant positive effect on OHISB in young and middle-aged patients with stroke.

H7: Hospitalization has a significant positive effect on OHISB in young and middle-aged patients with stroke.

H8: Treatment method has a significant positive effect on OHISB in young and middle-aged patients with stroke.

H9: Number of chronic diseases has a significant positive effect on OHISB in young and middle-aged patients with stroke.

**Psychological factors.**

**Perceived usefulness and perceived ease of use:** Solving related health problems and expanding health knowledge as well as the ease of using online health information affects individual attitudes towards using online health information, and thus affect OHISB. Li et al. found that both perceived ease of use and perceived usefulness positively affect

patient attitudes towards using online health information [18], and a Hong Kong study found that patient perceived ease of use can have a positive impact on OHISB [19]. Therefore, we proposed the following hypotheses:

H10: Perceived usefulness has a significant positive effect on OHISB in young and middle-aged patients with stroke.

H11: Perceived ease of use has a significant positive effect on OHISB in young and middle-aged patients with stroke.

**Self-efficacy:**　Self-efficacy can foster better information search strategies and learning outcomes in Internet-based environments [20]. Patients with high self-efficacy have a high willingness and frequency of using the Internet to search for health information [21]. Zhu et al. found that self-efficacy indirectly affects OHISB in a study on patients with Coronary Heart Disease [22]. Therefore, we proposed the following hypothesis:

H12: Self-efficacy has a significant positive effect on OHISB in young and middle-aged patients with stroke.

**Perceived benefit:**　Perceived benefit refers to the value perception that is beneficial to an individual after receiving the information. This study aimed to determine the extent to which OHISB can help young and middle-aged patients with stroke with their own health. Perceived benefits significantly affect patient OHISB [23]. Patients believe that OHISB can save time and relieve anxiety; therefore, they take the initiative to use it. Accordingly, the following hypothesis was proposed:

H13: Perceived benefit has a significant positive effect on OHISB in young and middle-aged patients with stroke.

**Perceived risk:**　Perceived risk refers to the negative consequences patients experience using OHISB. Yuchao found that perceived risk negatively affects patient trust, which in turn affects OHISB [24]. Based on this, the following hypothesis was proposed:

H14: Perceived risk has a significant negative effect on OHISB in young and middle-aged patients with stroke.

**Health anxiety:**　Health anxiety refers to the anxiety response caused by excessive worrying about one's health in the absence of health problems or pathological conditions [25]. Health anxiety prompts individuals to seek health information from various media sources. Son et al. conducted a survey on middle-aged people and found that both health concerns and health anxiety affect OHISB [26]. Therefore, we put forward the following hypothesis:

H15: Health anxiety has a significant positive effect on OHISB in young and middle-aged patients with stroke.

**Environmental factors.**
**Social influence:**　Social influence refers to the phenomenon that causes an individual's thoughts, feelings, and behaviors to change based on the actions of others. Studies have shown that when people who have a significant impact on patient beliefs search for online health information, or when hospitals and official social media promote online health care, the popularity of OHISB increases, and patients develop a strong intention to use and adopt this approach [27], so the following hypothesis was proposed:

H16: Social influence has a significant positive effect on OHISB in young and middle-aged patients with stroke.

**Information factors.**
**Information quality:**　Information quality includes the timeliness of information and the accuracy of content [28]. The quality of information affects the frequency of individual information searches, which determines the OHISB [29]. A systematic review by German scholars on the individual information needs of online family doctors showed that information quality is a key factor affecting the development of online information resources [30]. Thus, the following hypothesis was proposed:

H17: Information quality has a significant positive effect on OHISB in young and middle-aged patients with stroke.

**Privacy of information:** Individual health information is highly sensitive and requires considerable privacy. After COVID-19, college students have become worried about health information privacy leakage and have avoided online health information seeking [31]. Based on this, the following hypothesis was proposed:

H18: Information privacy has a significant negative effect on OHISB in young and middle-aged patients with stroke.

**Information technology factors.**

**E-health literacy:** E -health literacy refers to the ability of individuals to obtain the required health-related information from the Internet and apply it to solving health problems through understanding and judgment [32]. This is an important factor when using online health resources [33]. A qualitative study in Australia found that low e-health literacy was a prominent internal barrier to OHISB in patients with chronic diseases [34]. Accordingly, we proposed the following hypothesis:

H19: E-health literacy has a significant positive effect on OHISB in young and middle-aged patients with stroke.

## Materials and methods

### Setting and participants

This study was of a quantitative nature; research was conducted between October 31, 2023. Data were obtained from young and middle-aged patients with stroke receiving treatment at a Class III Grade A hospital in Shandong Province, China. Convenience sampling, a non-probability sampling method, was used. The sample size was determined according to Kendall's principle [35]. Generally, five to ten times the number of independent variables is the required sample size. This study included 19 independent variables. Considering a 10% invalid questionnaire recovery rate, a sample size of 106–212 was needed.

The inclusion criteria were as follows: meeting stroke diagnostic criteria [36], being aged 18–59 years, CT or MRI confirmation, having normal communication skills, being able to clearly express desired outcomes, and volunteering to participate in this research.

The exclusion criteria were as follows: having a mental illness or psychological disorder before admission, various major disease complications, having stroke along with a critical or life-threatening condition at any time, or having cognitive impairment.

The research staff screened participants according to the inclusion and exclusion criteria using the department's computer and medical records. The purpose and significance of the study were explained to the patients and their families, who were instructed to provide written informed consent and their contact information (E-mail address and telephone number). In the survey process, the investigators avoided inducing respondents to answer the questionnaire, so they could fill in the questionnaire independently. After completing the questionnaires, the investigators immediately collected the answered questionnaires and checked for gaps. Data collection durations lasted 30–40 min each. Qualified questionnaires were uniformly numbered and entered into Excel after double-checking. The researchers distributed 244 questionnaires and collected 230 valid responses. Patients were not paid to participate in the study, and the questionnaires were completed before hospital discharge.

### Measures

The questionnaire consisted of two parts. The first part was a general information questionnaire containing general demographic and disease data, and the second part was prepared according to previous studies. A tool was developed to measure the perceived usefulness of online health information among young and middle-aged patients with stroke [37]. The three analyzed items were "meeting health needs," "solving health problems, " and "saving time." A four-item credibility measurement instrument was used to measure perceived ease of use [38]. The four items were "easy to operate," "skilled use," "clear and easy to understand," and "easy to achieve what you want to do." The eight-item e-Health Literacy

Scale developed by Norman et al. measures a Cronbach's alpha of 0.94 [39]. Based on Hangkai's research, a three-item self-efficacy assessment tool was developed [40]. The perceived benefits were investigated according to Chaohua's research using three questions [41]. The Cronbach's alpha was 0.84 for the three items, which were "very convenient," "improve health," and "save time." A three-item credibility measurement tool was used to measure perceived risk, namely "uncertainty," "potential risk," and "greater risk.[42]" Health anxiety was measured using three items, and Cronbach's alpha was 0.78 [43]. A three-item information quality assessment tool was developed based on Xiaoling's research. The three items were "can provide reliable information," "can provide sufficient information," and "can provide timely information.[44]" Yuting et al. developed a three-item information privacy assessment tool [45]. Venkatesh et al. measured social influence using three questions [46]. According to the scale developed by Yoon and Kim, the frequency and intensity of OHISB are determined based on attention, browsing, querying, and utilization [47]. All the above scales passed the five-point Likert scoring method.

### Data analysis

After the questionnaires were collected, the data were imported into Excel spreadsheets (Microsoft Corp., Washington, Redmond). SPSS Statistics (version 23.0; IBM Corp., Armonk, NY, USA) was used for data analysis. Classification variables were described using frequency, and scores were described using mean ± standard deviation. Univariate analyses were performed using two independent sample t-tests or analysis of variance (ANOVA). Correlation analysis between all quantitative variables was performed using Pearson's (r) correlation coefficient. In the multivariate analysis, a multiple linear analysis was used to determine the factors influencing the OHISB scores ($P<0.05$).

### Ethical consideration

This study was approved by the Institutional Review Board of the Binzhou Medical University (IRB No. 2023–367). This study was approved and supported by the heads of nursing and related departments at the hospital. Written informed consent was obtained from all participants.

## Results

### Sample characteristics

The demographic information is presented in Table 1. Of the 230 respondents, 13.91% had a primary school education or below (n=32), 23.04% had a junior high school education (n=53), 33.91% had a high school or technical secondary school education (n=78), and the remaining 29.13% had a college education or above (n=67). Of these, 20.43% were farmers (n=47), 6.09% were students (n=14), 41.74% were full-time workers (n=96), 22.61% were freelance workers (n=52), 8.26% were laid-off or retired (n=19), and the remaining 0.87% had other jobs (n=2). The duration of stroke was less than 1 year in 47.83% (n=110), 1–5 years in 31.74% (n=73), 6–10 years in 12.61% (n=29), and >10 years in 7.83% of the patients (n=18). Of the patients, 70.50% were hospitalized once due to stroke (n=104), 34.78% were hospitalized 2–3 times (n=80), and the remaining 20% were hospitalized more than three times (n=46). Of the participants, 20% had no chronic diseases (n=46), 40.87% had one chronic disease (n=94), 26.52% had two chronic diseases (n=61), and 12.61% had three or more chronic diseases (n=29).

### Factors influencing OHISB in young and middle-aged patients with stroke

**Univariate analyses.** Table 1 shows the general information (gender, age, education level, monthly income, and occupation) of young and middle-aged patients with stroke and disease-related factors (stroke course, hospitalizations, treatment methods, and the number of combined chronic diseases) were used as independent variables. The total OHISB score was used as the dependent variable in the univariate analysis. No significant differences were observed between

**Table 1. Characteristics of young and middle-aged patients with stroke (n=230).**

| Variable | Category | n | Mean±SD | t/F | P |
|---|---|---|---|---|---|
| Sex | Men | 107 | 3.96±0.83 | 0.386 | 0.535 |
| | Women | 123 | 3.89±0.67 | | |
| Age | <44 | 70 | 4.29±6.57 | 5.260 | <0.001 |
| | ≥44 | 160 | 3.76±0.73 | | |
| Education | ≤Primary school | 32 | 3.31±0.77 | 32.558 | <0.001 |
| | Junior high school | 53 | 3.46±0.65 | | |
| | High school | 78 | 4.12±0.61 | | |
| | ≥College graduate | 67 | 4.35±0.55 | | |
| Occupational | Farmer | 47 | 4.01±0.61 | 0.465 | 0.802 |
| | Student | 14 | 4.05±0.80 | | |
| | worker | 96 | 3.92±0.73 | | |
| | Independent | 52 | 3.81±0.90 | | |
| | Being laid off or retired | 19 | 3.91±0.66 | | |
| | Other | 2 | 4.00±0.71 | | |
| Monthly income | <1000 | 17 | 3.87±0.62 | | |
| | 1000-2000 | 28 | 3.97±0.67 | | |
| | 2000–5000 | 91 | 3.96±0.77 | 0.472 | 0.789 |
| | 5000–10000 | 73 | 3.84±0.70 | | |
| | >10000 | 21 | 4.01±0.96 | | |
| Stroke course | < 1 year | 110 | 3.69±0.6 | 31.762 | <0.001 |
| | 1~5years | 73 | 3.76±0.73 | | |
| | 6~10years | 29 | 4.66±0.56 | | |
| | >10years | 18 | 4.81±0.41 | | |
| hospitalizations | 1 time | 104 | 3.58±0.61 | 57.297 | <0.001 |
| | 2~3times | 80 | 3.91±0.71 | | |
| | >3times | 46 | 4.73±0.35 | | |
| treatment methods | Surgery | 53 | 3.95±0.69 | 0.079 | 0.779 |
| | Conservative treatment | 177 | 3.92±0.76 | | |
| Number of chronic diseases | 0 species | 46 | 3.58±0.75 | 22.904 | <0.001 |
| | 1 species | 94 | 3.81±0.66 | | |
| | 2 species | 61 | 3.94±0.68 | | |
| | 3 or more | 29 | 4.81±0.39 | | |

men and women (t=0.386, $P$=0.535), different occupations (t=0.465, $P$=0.802), monthly income (t=0.472, $P$=0.789), or treatment methods (t=0.079, $P$=0.779). Significant differences were observed in age (t=0.386, $P$<0.001), educational level (t=32.558, $P$<0.001), stroke duration (t=31.762, $P$<0.001), number of hospitalizations due to stroke (t=57.297, $P$<0.001), and number of complications (t=22.904, $P$<0.001).

**Correlation analysis.** Table 2 shows the results of the correlation analysis. Perceived usefulness (r=0.612, $P$<0.01), perceived ease of use (r=0.553, $P$<0.01), e-health literacy (r=0.552, $P$<0.01), self-efficacy (r=0.515, $P$<0.01), perceived benefits (r=0.581, $P$<0.01), health anxiety (r=0.391, $P$<0.01), information quality (r=0.415, $P$<0.01) and social influence (r=0.335, $P$<0.01) were positively correlated with OHISB. Perceived risk (r=-0.226, $P$< 0.01) and information privacy (r=-0.285, $P$ < 0.01) were negatively correlated with OHISB.

**Multiple linear regression analyses.** Table 3 presents the results of the multiple linear regression model. This model examined variables that were statistically significant in the univariate and correlation analyses. The 15 independent

**Table 2. Correlation analysis of each dimension and OHISB in young and middle-aged patients with stroke(n=230).**

| Variable | OHISB |
|---|---|
| perceived ease of use | 0.612** |
| perceived usefulness | 0.553** |
| e-health literacy | 0.552** |
| self-efficacy | 0.515** |
| perceived benefits | 0.581** |
| perceived risk | -0.226** |
| health anxiety | 0.391** |
| information quality | 0.415** |
| information privacy | -0.285** |
| social influence | 0.335** |

variables explained 74.3% of the variance ($R^2$=0.743). The tolerance of the independent variables ranged 0.434–0.845, all of which were >0.2. The variance inflation factor was 1.183–2.225 (all < 5), indicating no multicollinearity between variables. Age (b=-0.128, t=-2.147, *P*=0.033), education level (b=0.074, t=2.266, *P*=0.024), stroke course (b=0.105, t=2.788, *P*=0.006), hospitalizations (b=0.120, t=2.685, *P*=0.008), number of chronic diseases (b=0.069, t=2.205, *P*=0.029), perceived ease of use (b=0.072, t=2.192, *P*=0.029), perceived usefulness (b=0.071, t=2.202, *P*=0.029), e-health literacy (b=0.084, t=2.235, *P*=0.026), self-efficacy (b=0.079, t=2.237, *P*=0.026), perceived benefits (b=0.109, t=3.068, *P*=0.002), health anxiety (b=0.090, t=2.700, *P*=0.007), information quality (b=0.072, t=2.416, *P*=0.017), social influence (b=0.061, t=2.156, *P*=0.032), perceived risk (b=-0.203, t=-7.254, *P*<0.001), and information privacy (b=-0.057, t=-2.064, *P*=0.040) were the influencing factors for OHISB in young and middle-aged patients with stroke (all *P*<0.05). Age, perceived risk, and information privacy were negatively correlated with OHISB scores.

**Table 3. Multiple linear regression analysis of OHISB in young and middle-aged patients with stroke(n=230).**

| Variable | b | Standard Error of B | Standar-dization coefficient β | t | P | Collinearity analysis Allowa-nce | VIF |
|---|---|---|---|---|---|---|---|
| (constant) | 1.928 | 0.214 | | 8.988 | <0.001 | | |
| Age | -0.128 | 0.060 | -0.079 | -2.147 | 0.033 | 0.826 | 1.210 |
| Education | 0.074 | 0.033 | 0.101 | 2.266 | 0.024 | 0.569 | 1.756 |
| Stroke course | 0.105 | 0.038 | 0.132 | 2.788 | 0.006 | 0.498 | 2.008 |
| hospitalizations | 0.120 | 0.045 | 0.123 | 2.685 | 0.008 | 0.532 | 1.881 |
| Number of chronic diseases | 0.069 | 0.032 | 0.087 | 2.205 | 0.029 | 0.719 | 1.391 |
| perceived ease of use | 0.072 | 0.033 | 0.111 | 2.192 | 0.029 | 0.434 | 2.306 |
| perceived usefulness | 0.071 | 0.032 | 0.106 | 2.202 | 0.029 | 0.480 | 2.083 |
| e-health literacy | 0.084 | 0.037 | 0.112 | 2.235 | 0.026 | 0.449 | 2.225 |
| self-efficacy | 0.079 | 0.035 | 0.100 | 2.237 | 0.026 | 0.558 | 1.792 |
| perceived benefits | 0.109 | 0.036 | 0.145 | 3.068 | 0.002 | 0.504 | 1.985 |
| perceived risk | -0.203 | 0.028 | -0.266 | -7.254 | <0.001 | 0.834 | 1.199 |
| health anxiety | 0.090 | 0.033 | 0.104 | 2.700 | 0.007 | 0.759 | 1.318 |
| information quality | 0.072 | 0.030 | 0.089 | 2.416 | 0.017 | 0.822 | 1.216 |
| information privacy | -0.057 | 0.028 | -0.075 | -2.064 | 0.040 | 0.845 | 1.183 |
| social influence | 0.061 | 0.028 | 0.079 | 2.156 | 0.032 | 0.825 | 1.213 |

## Discussion

### Current status of OHISB in young and middle-aged patients with stroke

The results of this study showed that the OHISB scores of young and middle-aged patients with stroke (15.69±2.98). According to the OHISB scale score interval (4–20 points), the scores of young and middle-aged patients with stroke was at a medium-high level. This may be related to the overall younger age of young and middle-aged patients with stroke, more experience in using smart devices, and greater ability to use electronic devices. At present, major medical institutions are developing online medical services one after another, and abundant easily accessed network resources provide the basic conditions for OHISB [48]. However, a lot of online information is present; therefore, screening for the correct and suitable information has become the top priority in OHISB. Therefore, patients should enhance their cognition and judgment of health-information resources to avoid being misled because of incorrect information. Online information publishers should strengthen platform services and internal supervision to ensure the authenticity of information and to improve the credibility of online health information. On this basis, relevant governmental departments should take measures to sort online health information resources and release more authoritative official health websites to promote OHISB among young and middle-aged patients with stroke.

### General demographic influencing factors

**Age and education level.** Through empirical analyses and demographic factors, such as age and educational level, can affect the OHISB in young and middle-aged patients with stroke, similar to the results reported by Chu et al. [49]. Regarding age, patients aged ≥44 years generally have lower OHISB scores than those aged <44 years. This may be because younger groups are usually better at using network technology to search for health information [50]. Younger groups are usually more familiar with new applications on the Internet and social media and can find the required information more quickly. By contrast, older groups may be slightly deficient in information search technologies, which may affect their ability to obtain online health information. Additionally, age may be related to patient health conditions and stages of disease recovery [51], thereby affecting the willingness and ability to search for and process online health information. Relevant departments should guide young people to improve their information-discrimination ability and use emerging technologies for personalized information search. Older patients can improve their efficiency in obtaining health information through skill training, family assistance, and teaching that combines traditional media and the Internet.

Regarding educational level, most young and middle-aged patients with stroke had a high school, college, or higher education; the educational level was generally high. Younger patients are usually more familiar with new Internet and social media applications and can quickly find the required information. Individuals with higher educational levels have a stronger ability to acquire and utilize health information and are willing to acquire disease-related knowledge through the Internet. Educational level can affect the ability of individuals to use this health knowledge. Individuals with higher education levels also have higher health literacy levels, are better able to judge the authenticity of online health information, and are more confident in using the health information. Medical institutions and relevant departments can establish communication platforms and facilitate expert interactions to guide patients in utilizing health information resources and should simplify online health information, provide regular offline guidance, and enable patients of different cultural backgrounds to access online health information resources.

### Disease factors

**Stroke course, number of hospitalizations, and number of chronic diseases.** In this study, the course of stroke, number of hospitalizations, and number of chronic diseases were significantly positively correlated with OHISB. The course of stroke is a crucial factor that influences OHISB. Disease duration may directly affect patient cognition of the disease and the demand for health information. According to one study, etiologies of ischemic stroke in young patients

are diverse. Complications may occur in patients, as the disease progresses [52]. This will increase the demand for information on disease management and the prevention of recurrence. Therefore, patients will be more motivated to search for online health information regarding the disease to manage their health conditions better. Researchers have observed that the OHISB in some patients decreases as the disease course progresses. The reason may be that a longer illness duration aggravates the disease burden, causing patients to experience negative emotions about the disease [53], reducing compliance and making them unwilling to spend time and energy to actively obtain health-related information. Medical personnel should offer targeted advice based on disease durations. For instance, some popular basic stroke science websites are recommended for patients with a shorter disease course. Some professional medical websites have been introduced for patients with a longer disease course. Medical staff should also increase the frequency of regular follow-up for patients and guide them in searching for online health information during the follow-up process.

The number of hospitalizations may reflect the severity and complexity of the stroke [54]. Patients with multiple hospitalizations already have a certain understanding of their disease conditions and related knowledge [55]. They may pay more attention to their health status or have various questions and confusions. Thus, they actively seek ways to understand diseases and gain treatment-related knowledge. Searching for relevant information on the Internet is one of the chosen methods [56]. Studies have shown that the number of hospitalizations due to stroke is one of the factors influencing the health information-seeking behavior in patients with stroke [57]. Medical staff should focus on introducing reliable sources of online health information for young and middle-aged patients, who are hospitalized for the first time.

The coexistence of multiple chronic diseases may increase treatment complexity. Patients should obtain information regarding drug interactions, lifestyle adjustments, and other aspects through online searches. In older patients with stroke, the number of coexisting chronic diseases affects their quality of life, which may indirectly affect their behavior in searching for health information [58]. Hospitals should establish multidisciplinary medical teams that provide comprehensive treatment and management suggestions for young and middle-aged patients with stroke and multiple chronic diseases. However, a multidisciplinary medical team is currently being established at our hospital.

### Psychological factors

**Perceived usefulness, perceived ease of use, self-efficacy, perceived benefits, health anxiety, and perceived risk.** According to the survey results, perceived usefulness, perceived ease of use, self-efficacy, perceived benefits, and health anxiety had significant positive impacts on OHISB in young and middle-aged patients with stroke. Schnall et al. concluded through focus group discussions that to promote user intentions to use mobile health technology, the technology needs to be perceived as useful and easy to use, with a relatively low perceived risk [59]. Young and middle-aged patients with stroke acquire relevant health information from the Internet. When they perceive that the information is useful and has a positive impact, they have a positive attitude towards OHISB, thereby promoting the utilization of online resources. Perceived ease of use is a prerequisite for OHISB in young and middle-aged patients with stroke. A clear and concise web page on an online health platform and smooth operations can enhance search intentions. This study found a significant negative correlation between perceived risk and OHISB. This is contrary to the results reported by Zhao et al. in patients with chronic diseases [60]. One possible reason for this is the differences in the degree of information demand. Young and middle-aged patients with stroke have urgent needs, pay more attention to usefulness, and consider risks less often. Patients with chronic diseases require long-term management and time to assess the risks. Based on the above analysis, relevant departments and network health information platforms should optimize platform design to ensure a simple interface and convenient operation and should provide multiple channels for information acquisition to enhance patient perceived ease of use. Review departments must guarantee the accuracy and authority of the information. They can collaborate with professional institutions and strictly review resources to boost patient perceived usefulness and reduce perceived risk.

Previous studies have shown that perceived benefits and self-efficacy play mediating roles in OHISB [61]. Self-efficacy serves as a driving force for continuous engagement in OHISB. Patients with high self-efficacy have greater confidence in disease management and are more inclined to obtain online health information that is highly timely and authoritative for maintaining their health [62], thereby enhancing the OHISB initiative. The manner in which perceived benefits exert an influence is similar to that in self-efficacy. Medical staff can improve patient self-efficacy by conducting lectures on network information. They can also enhance patient perceived benefits by presenting successful cases and clarifying rehabilitation expectations, thereby promoting OHISB.

Research has shown a positive correlation between health anxiety and OHISB [63]. Individuals with a higher level of health anxiety are more likely to frequently search for health information online, and this behavior may exacerbate their anxiety [64]. Health promoters should help patients manage health anxiety more effectively and concurrently improve health information literacy to avoid increased anxiety caused by excessive searching.

## Environmental factors

**Social influence.** Social influence is an important factor that influences OHISB in young and middle-aged patients with stroke [65]. In China, as their children are occupied with work, older patients with stroke tend to utilize online resources more frequently to acquire health information. Moreover, research indicates that these patients highly value peer support obtained through online platforms, such as WeChat [66]. Health promoters can enhance patient social influence by establishing patient communication groups and encouraging family participation, thereby promoting patients to search for and utilize health information more actively.

## Information factors

**Information quality and privacy.** Previous studies have found that information quality and privacy play important roles in seeking health aid through the Internet [67,68]. In this study, we found that information quality has a significant positive effect on OHISB, and information privacy has a significant negative effect on OHISB. If more advertisements are present on a website and the disclosure of patient privacy occurs frequently, even if the online resource is rich and easy to operate, the search intentions of young and middle-aged patients with stroke will significantly decrease. Wang et al. confirmed this view [69]. High-quality information and guaranteed privacy enables patients to trust online platforms, thereby increasing the willingness to seek health information through online channels. Therefore, establishing high-quality health information websites as well as strengthening and improving the design of health websites are the most important factors in promoting OHISB.

## Information technology factor

**E-health literacy.** This study found that e-health literacy had a significant positive effect on OHISB in young and middle-aged patients with stroke. Studies have shown that improving electronic health literacy in patients with stroke is extremely important [70]. The higher the level of e-health literacy, the stronger the ability to search for health information, and accordingly, the higher the quality of the health information obtained, and the more positive the attitude towards OHISB, which is consistent with the results reported by Cho et al. [71]. In clinical practice, the evaluation of e-health literacy in young and middle-aged patients with stroke should be strengthened. Medical institutions should conduct targeted publicity and educational activities to improve literacy and guide correct OHISB.

## Limitations

First, this was a cross-sectional study, and the causal interpretation of OHISB for each variable was limited but can be further clarified in future longitudinal studies. Second, the size and representatives of the sample, drawn from only one

hospital, were limited. Therefore, caution should be exercised in standardizing populations that do not match the characteristics of the sample. Third, some unknown factors also had an impact on OHISB in young and middle-aged patients with stroke.

## Conclusion

This study explored the current status of OHISB in young and middle-aged patients with stroke and identified the factors affecting OHISB. Although OHISB in young and middle-aged patients with stroke is at a medium-to-high level, many problems remain to be solved. For example, some patients have deviations in their understanding of disease-related knowledge, whereas others cannot effectively apply the obtained information to self-health management after obtaining information. Healthcare professionals need to develop appropriate interventions for young and middle-aged patients with stroke based on modifiable influencing factors.

## Supporting information

**S1 Data. Original data.**
(XLSX)

## Acknowledgments

The authors are grateful to the participants for their time and effort, and Jiaming Zhang for his help with data collection. Jun Zhao and Zhen Peng contributed to the study design and data analysis. Ge Shi wrote the first version of the manuscript. Li Shang and Jiajia Yu proofread the manuscript, provided comments, and revised it, and all authors approved the final version.

## Author contributions

**Conceptualization:** Ge Shi, Jiajia Yu, Li Shang.

**Data curation:** Ge Shi, Jiajia Yu, Li Shang, Jiaming Zhang, Jun Zhao, Zhen Peng.

**Formal analysis:** Ge Shi, Li Shang, Jiaming Zhang.

**Investigation:** Ge Shi, Jiajia Yu, Li Shang, Jiaming Zhang, Jun Zhao, Zhen Peng.

**Methodology:** Ge Shi, Li Shang.

**Supervision:** Li Shang.

**Writing – original draft:** Ge Shi.

**Writing – review & editing:** Jiajia Yu, Li Shang.

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
