## [Decision Letter · Decision Letter 0]

8 Oct 2024

PONE-D-24-34419Factors affecting online health information seeking behaviour in young and middle-aged stroke patientsPLOS ONE

Dear Dr. Shang,

Thank you for submitting your manuscript to PLOS ONE. After careful consideration, we feel that it has merit but does not fully meet PLOS ONE’s publication criteria as it currently stands. Therefore, we invite you to submit a revised version of the manuscript that addresses the points raised during the review process.

We look forward to receiving your revised manuscript.

Kind regards,

Ali Garavand

Academic Editor

PLOS ONE

Journal Requirements: When submitting your revision, we need you to address these additional requirements. 1. Please ensure that your manuscript meets PLOS ONE's style requirements, including those for file naming. The PLOS ONE style templates can be found at https://journals.plos.org/plosone/s/file?id=wjVg/PLOSOne_formatting_sample_main_body.pdf and https://journals.plos.org/plosone/s/file?id=ba62/PLOSOne_formatting_sample_title_authors_affiliations.pdf 2. PLOS requires an ORCID iD for the corresponding author in Editorial Manager on papers submitted after December 6th, 2016. Please ensure that you have an ORCID iD and that it is validated in Editorial Manager. To do this, go to ‘Update my Information’ (in the upper left-hand corner of the main menu), and click on the Fetch/Validate link next to the ORCID field. This will take you to the ORCID site and allow you to create a new iD or authenticate a pre-existing iD in Editorial Manager. 3. Please review your reference list to ensure that it is complete and correct. If you have cited papers that have been retracted, please include the rationale for doing so in the manuscript text, or remove these references and replace them with relevant current references. Any changes to the reference list should be mentioned in the rebuttal letter that accompanies your revised manuscript. If you need to cite a retracted article, indicate the article’s retracted status in the References list and also include a citation and full reference for the retraction notice.

Reviewers' comments:

Reviewer's Responses to Questions

**Comments to the Author**

1. Is the manuscript technically sound, and do the data support the conclusions?

Reviewer #1: Yes

Reviewer #2: Yes

2. Has the statistical analysis been performed appropriately and rigorously? 

Reviewer #1: I Don't Know

Reviewer #2: No

3. Have the authors made all data underlying the findings in their manuscript fully available?

Reviewer #1: Yes

Reviewer #2: Yes

4. Is the manuscript presented in an intelligible fashion and written in standard English?

Reviewer #1: No

Reviewer #2: Yes

5. Review Comments to the Author

Reviewer #1: Dear author, this article contributes valuable knowledge to the field of health information behavior, particularly in the context of stroke patients, and offers practical insights for enhancing online health information literacy and patient engagement. The comments provided will help improve your article.

Abbreviations should not be used in the abstract and the methodology should be presented in more detail. Include keywords related to research.

The introduction is very long and needs to be presented more densely.

The methodology should be presented in more detail. The sample volume formula should be provided. Mention the inclusion and exclusion criteria exactly.

Discussion emphasizes positive findings and practical implications, while potential negative aspects or challenges are less explored.

Some references appear to be cited multiple times (e.g., Deng Chaohua's studies).

There are inconsistencies in the citation style.

Ensure proper spacing and punctuation.

Improve sentence structure for clarity.

Reviewer #2: Dear Authors of interesting manuscript. I appreciate an opportunity to review your paper. Please find my comments and suggestions below.

1. In line 1 abstract, you must state the complete phrase with its abbreviation, and later you can use only the abbreviation. You must write online health information seeking behavior (OHISB) for the first time.

2. In introduction section, the statistics used are not up to date. (references 4 and 5)

3. In Theoretical development and hypotheses section, paragraph about “the second Wilson model” has no reference.

4. The methodology section lacks detailed explanations of the data collection and statistical analysis methods. Provide more comprehensive details about the sampling process, data collection instruments, and specific statistical methods used.

5. In discussion section, some sentences haven’t any references for example in Social influence you say: Studies have shown that social influence can significantly improve OHISB

in stroke patients. Which studies?

Also, in this section, you include more comparisons with previous studies, highlighting how your findings align or contrast with them.

6. PLOS authors have the option to publish the peer review history of their article (what does this mean? ). If published, this will include your full peer review and any attached files.

**Do you want your identity to be public for this peer review?** For information about this choice, including consent withdrawal, please see our Privacy Policy .

Reviewer #1: No

Reviewer #2: No

---

## [Author Response · Author response to Decision Letter 1]

18 Oct 2024

Reviewers' comments:

Reviewer #1:

1.Abbreviations should not be used in the abstract and the methodology should be presented in more detail. Include keywords related to research.

Thanks to the reviewers for their meticulous review and valuable suggestions. Modifications have been made as suggested. In the modification, we removed the abbreviation "OHISB" and fully supplemented it as "online health information seeking behaviour". We supplemented the methodology (lines10-15), including the sampling method, the experimental time period, and the basis for including the dimensions of influencing factors. Keywords were added below the abstract.

2.The introduction is very long and needs to be presented more densely.

Thanks to the reviewers for their meticulous review and valuable suggestions. Modifications have been made as suggested. We have revised the logic and language of the introduction. The data on stroke epidemiology(lines31-33) and online health information seeking behavior(lines45-50) have been replaced with data from other countries that are more representative.

3.The methodology should be presented in more detail. The sample volume formula should be provided. Mention the inclusion and exclusion criteria exactly.

Thanks to the reviewers for their meticulous review and valuable suggestions. Modifications have been made as suggested. We have supplemented the method section in detail, including the sampling method(lines196-199), the sample volume formula(lines199-203), inclusion and exclusion criteria(lines203-209), data collection(lines244-246), and statistical analysis(lines246-250).

4.Discussion emphasizes positive findings and practical implications, while potential negative aspects or challenges are less explored.

Thanks to the reviewers for their meticulous review and valuable suggestions. Modifications have been made as suggested. We have supplemented the negative impacts and current challenges faced in the parts of stroke course(lines368-371), number of combined chronic diseases(lines391-394), and perceived risk(lines407-413).

5.Some references appear to be cited multiple times.There are inconsistencies in the citation style.Ensure proper spacing and punctuation.Improve sentence structure for clarity.

Thanks to the reviewers for their meticulous review and valuable suggestions. Modifications have been made as suggested.We have removed duplicate citations, such as Deng Chaohua(lines146-147),Zhao YC(lines140-141) and Lee(lines456-457).

Additionally, we have checked the full-text citations and sentences.

Reviewer #2:

1.In line 1 abstract, you must state the complete phrase with its abbreviation, and later you can use only the abbreviation. You must write online health information seeking behavior (OHISB) for the first time.

Thanks to the reviewers for their meticulous review and valuable suggestions. Modifications have been made as suggested. In the modification, we removed the abbreviation "OHISB" and fully supplemented it as "online health information seeking behaviour".

2.In introduction section, the statistics used are not up to date. (references 4 and 5)

Thanks to the reviewers for their meticulous review and valuable suggestions. Modifications have been made as suggested. We have updated the data of stroke patients(lines31-33) and young and middle-aged stroke patients(lines34-36).

3.In Theoretical development and hypotheses section, paragraph about “the second Wilson model” has no reference.

Thanks to the reviewers for their meticulous review and valuable suggestions. Modifications have been made as suggested. We have supplemented the references of the second Wilson model, which are Reference 13 and Reference 14 respectively.

4.The methodology section lacks detailed explanations of the data collection and statistical analysis methods. Provide more comprehensive details about the sampling process, data collection instruments, and specific statistical methods used.

Thanks to the reviewers for their meticulous review and valuable suggestions. Modifications have been made as suggested. We have supplemented the content of data collection and statistical analysis methods, including the sampling process(lines196-209), data collection tools(lines244-246) and statistical methods(lines246-250).

5.In discussion section, some sentences haven’t any references for example in Social influence you say: Studies have shown that social influence can significantly improve OHISB in stroke patients. Which studies?Also, in this section, you include more comparisons with previous studies, highlighting how your findings align or contrast with them.

Thanks to the reviewers for their meticulous review and valuable suggestions. Modifications have been made as suggested. We have supplemented the citation of references in the discussion section. Moreover, in the discussion section, we have comprehensively added the comparison between the research results of this study and previous studies, such as in the perception of perceived risk(lines407-418).

---

## [Decision Letter · Decision Letter 1]

3 Mar 2025

PONE-D-24-34419R1Factors affecting online health information seeking behavior in young and middle-aged stroke patientsPLOS ONE

Dear Dr. Shang,

Thank you for submitting your manuscript to PLOS ONE. After careful consideration, we feel that it has merit but does not fully meet PLOS ONE’s publication criteria as it currently stands. Therefore, we invite you to submit a revised version of the manuscript that addresses the points raised during the review process.

**ACADEMIC EDITOR:** Dear authors,

As you will see below, both reviewers are satisfied with the revisions and feel that you have addressed their concerns and suggestions appropriately. However, one reviewer noted that the manuscript requires extensive language polishing before it can be accepted for publication.

I therefore ask you to have the entire manuscript reviewed by a native English speaker. I look forward to receiving the revised version.

Best wishes

Nicola Diviani

We look forward to receiving your revised manuscript.

Kind regards,

Nicola Diviani

Academic Editor

PLOS ONE

Journal Requirements:

Reviewers' comments:

Reviewer's Responses to Questions

**Comments to the Author**

1. If the authors have adequately addressed your comments raised in a previous round of review and you feel that this manuscript is now acceptable for publication, you may indicate that here to bypass the “Comments to the Author” section, enter your conflict of interest statement in the “Confidential to Editor” section, and submit your "Accept" recommendation.

Reviewer #1: All comments have been addressed

Reviewer #2: All comments have been addressed

2. Is the manuscript technically sound, and do the data support the conclusions?

Reviewer #1: Yes

Reviewer #2: Yes

3. Has the statistical analysis been performed appropriately and rigorously?

Reviewer #1: I Don't Know

Reviewer #2: Yes

4. Have the authors made all data underlying the findings in their manuscript fully available?

Reviewer #1: Yes

Reviewer #2: Yes

5. Is the manuscript presented in an intelligible fashion and written in standard English?

Reviewer #1: No

Reviewer #2: Yes

6. Review Comments to the Author

Reviewer #1: Dear Authors,

The reviewers' comments have been incorporated into the manuscript. However, the article requires significant revisions to improve the language and grammar.

Reviewer #2: (No Response)

7. PLOS authors have the option to publish the peer review history of their article (what does this mean? ). If published, this will include your full peer review and any attached files.

**Do you want your identity to be public for this peer review?** For information about this choice, including consent withdrawal, please see our Privacy Policy .

Reviewer #1: No

Reviewer #2: No

---

## [Author Response · Author response to Decision Letter 2]

9 Mar 2025

Thanks to the reviewers for their meticulous review and valuable suggestions. We have had the manuscript professionally edited for language by Editageto ensure clarity and adherence to academic writing standards. A certificate of editing has been attached for your reference.

---

## [Editor Report · Decision Letter 2]

11 Mar 2025

Factors affecting online health information-seeking behavior in young and middle-aged patients with stroke

PONE-D-24-34419R2

Dear Dr. Shang,

We’re pleased to inform you that your manuscript has been judged scientifically suitable for publication and will be formally accepted for publication once it meets all outstanding technical requirements.

Kind regards,

Nicola Diviani

Academic Editor

PLOS ONE
---

## [Editor Report · Acceptance letter]

PONE-D-24-34419R2

PLOS ONE

Dear Dr. Shang,

I'm pleased to inform you that your manuscript has been deemed suitable for publication in PLOS ONE. Congratulations! Your manuscript is now being handed over to our production team.

Kind regards,

on behalf of

Dr. Nicola Diviani

Academic Editor

PLOS ONE